# Effects of Locomotion Regulatory Mode on COVID-19 Anxiety: The Mediating Role of Resilience

**DOI:** 10.3390/ijerph20156533

**Published:** 2023-08-06

**Authors:** Calogero Lo Destro, Alberto Costa

**Affiliations:** Department of Psychology, Niccolò Cusano University, 00166 Rome, Italy; alberto.costa@unicusano.it

**Keywords:** locomotion, regulatory modes, resilience, COVID-19, COVID-19 anxiety

## Abstract

The COVID-19 pandemic has exerted a significant impact on mental health globally. The uncertainty, fear, and stress associated with this crisis have contributed to a heightened prevalence of anxiety, depression, and various other mental health disorders. In this scenario, the present study aimed at investigating the relationship between locomotion regulatory mode, resilience, and COVID-19 anxiety. It is worth noting that previous extensive research has established a significant correlation between high levels of locomotion and diverse positive psychological conditions, such as optimism, reduced hopelessness, and a positive effect. A total of 243 participants completed measures of locomotion regulatory mode, resilience, and COVID-19 anxiety. In line with our hypotheses, individuals’ locomotion regulatory mode was negatively, although non-significatively, associated with COVID-19 anxiety. Furthermore, resilience was found to mediate the relationship between the locomotion regulatory mode and COVID-19 anxiety, indicating that individuals displaying high locomotion may be better equipped to cope with the stress and uncertainty of the COVID-19 pandemic due to their greater levels of resilience. Taken together, these findings highlight the importance of considering both locomotion and resilience in managing anxiety related to COVID-19, and suggest that interventions aimed at enhancing resilience may be particularly beneficial for individuals with a low locomotion regulatory mode.

## 1. Introduction

The coronavirus infection that first emerged in Wuhan, China, is responsible for the respiratory illness known as COVID-19, which has quickly spread across the globe.

As of 19 June 2023, according to the World Health Organization official website, 767,984,989 confirmed COVID-19 cases have been reported [1]. In Italy, the total number of cases registered as positive is more than 25 million, with 190,706 deaths [2].

Italy was the first country in Europe to experience the impact of COVID-19 and has had one of the highest clinical burdens compared to other countries [3].

In March 2020, Italy was also the first country to enforce a nation-wide stay-at-home order to mitigate the spread of the virus. This lockdown confined over 60 million people within their homes for almost three months.

The COVID-19 pandemic and the subsequent implementation of lockdown measures have exerted a significant influence on the mental health and psychological well-being of individuals, not only within Italy, but also across the globe. In this vein, the literature [4] has widely documented a worldwide increase in the incidence of psychiatric disorders. Specifically, the COVID-19 pandemic has profoundly affected individuals’ psychological well-being, leading to an increase in anxiety, fear and distress [5,6,7]. The pandemic has disrupted individuals’ daily routines, resulting in emotions of powerlessness and seclusion [8]. Multiple studies [9,10] have demonstrated that the fear of contracting the virus, the loss of loved ones, and financial instability were the most contributing factors to elevated levels of psychological distress.

Therefore, the heightened levels of concern for personal health and economic consequences, along with increased levels of stress and a reduction in alternative activities, have significantly affected people’s lifestyle habits.

### 1.1. COVID-19 Anxiety Predictors

Although the long-term effects of the pandemic on mental health are yet to be fully understood, it has become increasingly clear that the psychological aftermath of COVID-19 will continue to be a significant public health issue for years to come.

For this reason, it appears crucial to ascertain primary factors involved in the development of anxiety related to COVID-19. Research [11] has shown that perceived threat of the virus, encompassing apprehensions regarding virus contraction, the gravity of the illness and the potential repercussions at both individual and societal levels, indeed play a significant role in this regard. Other predictors include financial concerns, social isolation, loneliness, pre-existing mental health conditions, and exposure to misinformation or conflicting information about the pandemic [12,13,14]. Additionally, demographic factors such as age, gender, and ethnicity have been found to influence the level of anxiety experienced during the pandemic [15,16].

On the other hand, psychological resilience, coping behaviors, and social support have become crucial tools in managing COVID-19 impacts on mental health in healthcare workers [17,18], hospitalized patients [19], and the general population [20].

In recent studies conducted during the COVID-19 pandemic, researchers of [21] have revealed a negative association between resilience and anxiety among physicians. Similar findings have been obtained in a Chinese sample of patients experiencing COVID-19 symptoms [22]. Specifically, these results indicated that resilience was inversely related with anxiety, serving as a protective factor against its development. Similarly, among the general population, it has been found that resilience negatively predicted COVID-19 anxiety and this relationship was mediated by persistent dysfunctional thinking about COVID-19 [23]. Furthermore, it has been shown [24] that individuals exhibiting higher levels of resilience, as opposed to those with lower levels, demonstrated a reduced susceptibility to the perceived threat of COVID-19, resulting in lower levels of future anxiety.

### 1.2. Resilience and Anxiety

The aforementioned findings are aligned with preexisting literature on the topic. In particular, research [25,26] has demonstrated that individuals who display high levels of resilience tend to have lower levels of anxiety, as they are better able to cope with stress and challenges. Conversely, those with lower levels of resilience tend to experience higher levels of anxiety in response to stressors. Resilience emerged as a mediator in associations between agreeableness, openness, conscientiousness, and anxiety symptoms in a sample of medical students [27]. Specifically, individuals who obtained higher scores in agreeableness, conscientiousness, and openness demonstrated elevated levels of resilience, which, in turn, corresponded to reduced anxiety symptoms. Conversely, higher scores in neuroticism among the students were linked to lower levels of resilience, subsequently leading to heightened anxiety symptoms. In another research endeavor [28], the role of specific resilience subfactors, namely positive thinking, tenacity, and help-seeking, was explored. The study utilized a sample of adolescents and revealed that these resilience factors acted as mediators between approach-coping and psychopathology. Specifically, both positive thinking and help-seeking were identified as mediators in the link between approach-coping and internalizing disorders such as anxiety and depression. The findings of another study [29], conducted with a sample of lung cancer patients, revealed that resilience acted as a partial mediator in the relationship between the perceived subjective support and anxiety levels. Moreover, resilience entirely mediated the relationship between patient support utilization and anxiety levels. Therefore, resilience seems to serve as a protective factor against anxiety and other negative mental health outcomes [30,31] and interventions aimed at increasing resilience may be effective in reducing anxiety symptoms [32].

### 1.3. Locomotion Regulatory Mode

Taken together, these findings highlight the extensive research conducted to examine the influence of risk and protective factors in relation to psychological distress and anxiety associated with the pandemic. Nevertheless, the role of regulatory modes in this context remains largely unexplored to date.

Based on the regulatory mode theory proposed by [33], there are two regulatory mode orientations, namely assessment and locomotion. The present paper primarily focuses on the locomotion mode, which is defined as the self-regulatory aspect that involves transitioning from one state to another and directing psychological resources toward initiating and maintaining progress toward goals with minimal interference or delay. Individuals with high levels of locomotion mode are inclined to prioritize rapid action and uninterrupted movement, rather than engaging in critical appraisal, as suggested by [34].

Research has established a positive association between high levels of locomotion and various positive psychological states, including optimism, increased self-esteem, achievement orientation, positive affect, and psychological well-being [33,35]. High levels of locomotion are also associated with increased job satisfaction, work engagement, and workaholism [36,37,38,39].

On the other hand, locomotion has also been negatively associated with social anxiety and depression [33]. In addition, it has been demonstrated that individuals with high levels of locomotion tend to exhibit lower levels of hopelessness, psychological strain, stress, burnout, turnover intentions, and withdrawal behaviors [37,40,41].

Furthermore, locomotion has been found to be positively associated with harmonious passion, which, in turn, is linked to better psychological adjustment and less stress in the workplace [42]. Likewise, within the realm of sports, the locomotion regulatory mode has been identified as a predictor of harmonious passion, leading to a subsequent reduction in athletes’ stress levels [43].

Finally, when examining the predominance of the regulatory mode (determined by subtracting assessment from locomotion scores), it has been observed as a predictive factor for work-related stress in a two-wave longitudinal design [44].

### 1.4. Locomotion and Resilience

Locomotion can be considered a significant empowering element and predictor of positive affect that supports the resilience process [45]. This association can be explained by the inclination of individuals with such regulatory mode to prioritize progress and movement, consequently reducing their propensity to ruminate on negative aspects of their past or present circumstances [46].

Moreover, research has identified locomotion orientation as a significant predictor [47] of resilient reintegration, which refers to the coping process involving personal growth, self-awareness, and the development of resilient characteristics. In this vein, it has been proposed that when recovery resources are inadequate for the demands of the situation, a negative cycle of increased anxiety levels may occur, causing a rupture of the body’s homeostasis [48].

Based on previous scientific literature emphasizing internal resources and energy of individuals exhibiting high levels of locomotion [33,43,49,50], it is suggested that resilience is fostered through the resources of individuals who are proactive and maintain a steadfast focus on their goals. Therefore, this mechanism is expected to effectively contribute to preventing the escalation of anxiety.

### 1.5. Research Hypotheses

In line with these notions, the present research aims to address the gap in the existing literature concerning the role of regulatory modes, particularly locomotion, in relation to COVID-19 anxiety and resilience. Despite the growing body of research on psychological responses to the COVID-19 pandemic, limited attention has been given to the influence of locomotion regulatory mode on individuals’ anxiety levels and their ability to cope resiliently with pandemic-related stressors. This gap is crucial to fill as understanding how the locomotion mode affects individuals’ responses to challenging circumstances like a pandemic can provide valuable insights into effective coping strategies and mental well-being promotion. Thus, this work can contribute to shedding light on the psychological mechanisms at play during unprecedented global events and potentially informing targeted interventions to bolster individuals’ adaptive responses and overall well-being.

The hypotheses of the present work posit that the locomotion regulatory mode, characterized by its emphasis on goal-directed action and progress, may influence individuals’ ability to cope with challenges and reduce anxiety related to the pandemic. Specifically, locomotion is expected to be positively related with resilience, whereas it should be negatively associated with COVID-19 anxiety. Furthermore, it is hypothesized that the locomotion effect on COVID-19 anxiety could be mediated by individuals’ resilience. Particularly, we suggest that this relationship between the locomotion mode and COVID-19 anxiety is contingent upon the level of resilience individuals possess, with ahigher level of resilience serving as a protective factor against anxiety.

## 2. Materials and Methods

### 2.1. Participants

Two hundred and forty-three participants (44 males) were recruited. We employed a snowball sampling strategy with the aim of recruiting individuals from the general population residing in Italy during the COVID-19 epidemic. An online survey was distributed across various provinces of Italy to gather data. No exclusion criteria were applied during participant selection. Participants were thoroughly informed about the study and provided informed consent for the anonymous use of their data. They completed a web-mediated survey on a voluntary basis and did not receive any compensation in exchange for their participation. Data were collected through Google Forms, with the questionnaire duration approximately lasting 10 min. The anonymity of the participants was ensured by not collecting any personally identifiable information during the questionnaire, thus safeguarding their privacy and confidentiality. The study was conducted in accordance with the Declaration of Helsinki.

The original sample consisted of 262 participants, but 19 of them were excluded from further analysis as they failed to provide a correct response to a control question. Participants’ mean age was 31.24 (*SD* = 10.92). Of the participants, 6.6% of held a post-graduate specialization, 37.9% held a university degree, 55.1% held a high school degree and 0.4% held a middle school diploma. Th sample characteristics are reported in Table 1.

### 2.2. Procedure and Materials

The survey consisted of several questions about socio-demographic data. This was followed by a second section concerning COVID-19 information, locomotion regulatory mode and resilience. Lastly, COVID-19 anxiety was measured. 

*COVID-19 Vaccination*. Participants had to respond to a question regarding their COVID-19 vaccination status. They were required to indicate whether they had received a COVID-19 vaccine by selecting either the response option “yes” or “no”.

*COVID-19 Infection*. Participants were instructed to respond to an inquiry regarding COVID-19 infection, in which they were required to select one of three options: “yes”, “no”, or “do not know” to indicate whether they had contracted COVID-19 in the past month. Subsequently, to facilitate additional analyses, the responses “no” and “do not know” have been combined.

*Locomotion Scale*. The Italian version of this scale, as designed by [33], contains 12 self-reported items aimed at evaluating the differences in locomotion tendency among individuals. Specifically, respondents rate the extent to which they agree with self-descriptive statements, both in positive (e.g., “I enjoy actively doing things, more than just watching and observing.”) and negative wording (e.g., “When I finish one project, I often wait a while before getting started on a new one.”). The responses were recorded on a 6-point Likert scale, ranging from “strongly disagree” (1) to “strongly agree” (6). The negative statements were reversed before analysis and a final score was determined by taking the average of all responses. Prior research, including Italian samples, has shown satisfactory temporal stability and reliability of the locomotion scale (Cronbach’s α = 0.82 for the locomotion scale and 0.78 for the assessment scale; [33]). In this sample, Cronbach’s α for the locomotion scale was 0.80 (*M* = 4.74, *SD* = 0.63).

*Resilience*. The Brief Resilience Scale [51] consists of six items that are designed to assess an individual’s level of resilience. The scale features three positively worded items and three negatively worded items. An example of a positive item is “I tend to bounce back quickly after hard times”. Conversely, an example of a negative item is “I have a hard time making it through stressful events”. Participants were asked to rate their agreement with each of the six statements using a 5-point Likert scale, ranging from 1 (strongly disagree) to 5 (strongly agree). In the present sample, Cronbach’s α was 0.87 for the resilience scale (*M* = 3.15, *SD* = 0.81).

*Covid anxiety scale*. COVID-19 anxiety scale [52] was developed to assess COVID-19-related anxiety. The scale is composed of seven items designed to assess individual differences. Specifically, participants had to rate the extent to which each item (e.g., “I feel heart racing when I read about COVID-19”) reflected their behavior in the last days. The responses were recorded on a 4-point Likert scale, ranging from 0 (not applicable to me) to 3 (very applicable to me). In this sample, Cronbach’s α was 0.92 (*M* = 0.66, *SD* = 0.67).

## 3. Results

Out of the total 243 participants, 13 reported that they had not received anti-COVID-19 vaccination. Additionally, only eight participants reported with certainty that they had contracted COVID-19 within the past 30 days. Two T-tests were conducted to examine potential differences in COVID-19 anxiety. The first T-test compared anxiety levels between individuals who chose not to get vaccinated (M = 0.27, SD = 0.43) and those who received the vaccine (M = 0.68, SD = 0.68). The results indicated that those who decided not to get vaccinated reported significantly lower anxiety levels, t (241) = −2.13, *p* = 0.01.

Additionally, the second T-test examined anxiety levels among individuals who had contracted COVID-19 in the past 30 days (M = 1.04, SD = 1.16) compared to those who had not (M = 0.64, SD = 0.65). The findings revealed a significant effect, with those who recently contracted the virus experiencing higher anxiety levels than those who did not, t (241) = −1.63, *p* < 0.001.

### 3.1. Correlation Analyses

Table 2 presents correlations between the study variables. Analyzing sociodemographic and COVID-19 related variables, it can be seen that gender (dummy-coded as Female = 0 and Male = 1) was significantly and negatively correlated with both locomotion and COVID-19 anxiety. These correlations indicated that females exhibited a higher level of COVID-19 anxiety and locomotion in comparison to males. Age was positively associated with the education level reported, which is reasonable given the sample’s age distribution, with many participants (i.e., 102) falling within the 18–25 age range, likely pursuing education. Being older was also positively associated with both higher levels of locomotion and resilience. On the other hand, younger people reported higher levels of COVID-19 anxiety. They also reported being infected within the past 30 days (dummy-coded as No and Do not Know = 0 and Yes = 1), which was positively associated with higher levels of education and having received COVID-19 vaccination (dummy-coded as No = 0 and Yes = 1). Interestingly, having received COVID-19 vaccination was positively associated with COVID-19 anxiety. These associations will be further explored in the discussion section.

More importantly, focusing on the main variable of the present study, a positive and significant correlation between locomotion and resilience was found, such that people reporting higher levels of locomotion, whereby participants described themselves as more resilient. On the other hand, COVID-19 anxiety was negatively and significantly related with resilience, such that more resilient individuals also displayed less COVID-19 anxiety.

### 3.2. Regression Model

To test the hypothesis that resilience mediated the relationship between locomotion and COVID-19 anxiety, a conditional process modeling program, PROCESS, was used. Specifically, a regression-based mediation analysis was performed with SPSS PROCESS macro Model 4 [53], which allows for both simple and parallel mediations and utilizes the bootstrapping method to extrapolate estimates of direct and indirect effects. Particularly, indirect effects were subjected to follow-up bootstrap analyses with 1000 bootstrap samples and 95% bias-corrected confidence intervals. Following the recommendation of [54], predictor variables were centered. Gender, age, education, COVID-19 vaccination and COVID-19 infection were entered as control variables. A summary of the results of these analyses is reported in Table 3.

First, the results showed that gender had a significant and positive effect on resilience (*B* = 0.27, *SE* = 0.13, *p* = 0.035), whereas it had a significant and negative effect on COVID-19 anxiety (*B* = −0.33, *SE* = 0.11, *p* = 0.003). Specifically, these results revealed that males reported higher levels of resilience compared to females; on the other hand, females reported a greater level of COVID-19 anxiety. Age was significantly and positively associated with resilience (*B* = 0.01, *SE* = 0.00, *p* = 0.002), such that older people reported being more resilient compared to younger people. Focusing on control variables closely related to COVID-19 contraction, it was found that receiving COVID-19 vaccination was associated with higher COVID-19 anxiety reports (*B* = 0.48, *SE* = 0.19, *p* < 0.011).

More importantly, the results of the regression-based mediation analysis revealed that locomotion had a positive and significant effect (*B* = 0.38, *SE* = 0.08, *p* < 0.001) on resilience, which, in turn, had a significant and negative effect on COVID-19 anxiety (*B* = −0.12, *SE* = 0.06, *p* = 0.038). Furthermore, the association between locomotion and COVID-19 anxiety was non-significant when controlling for resilience. Such results seem to indicate that the relationship between locomotion regulatory mode and COVID-19 anxiety is fully mediated by resilience (See Figure 1). Accordingly, the indirect effect of locomotion on COVID-19 anxiety through resilience was significant (*B* = −0.04, *BootSE* = 0.02; bootstrapping CI = [−0.09, −0.01]).

## 4. Discussion

The COVID-19 pandemic has dramatically changed people’s lives worldwide, causing unique levels of anxiety and stress. The long-lasting effects of the virus and its related stressors have made it challenging for individuals to maintain effective coping strategies, leading to persistent psychological challenges for many. In response to such crisis, researchers have been working to investigate the main factors influencing people’s ability to be resilient while facing COVID-19 emergency. In the same vein, understanding COVID-19 anxiety predictors seems essential for identifying those most at risk, developing effective interventions to reduce anxiety symptoms and promoting resilience during this challenging time.

In this regard, the present study brings novelty and unique insights to the forefront by investigating the role of the locomotion regulatory mode within the context of COVID-19 anxiety. Understanding how individuals adopt and utilize locomotion-oriented coping mechanisms during this unprecedented pandemic is of utmost importance, as it sheds light on adaptive responses and resilience-building strategies. Specifically, this research aimed at examining the role of the locomotion regulatory mode in the relationship between resilience and COVID-19 anxiety. In line with the hypotheses, this research demonstrated that locomotion regulatory mode was negatively, although not directly, related to COVID-19 anxiety and positively associated with resilience. This suggests that people with a goal-oriented approach to life are more able to face challenging circumstances and less likely to experience anxiety related to the pandemic. Additionally, the study found that resilience was negatively associated with COVID-19 anxiety. This implies that people who are more resilient are better able to cope with anxiety related to the pandemic.

In the present study, we additionally identified that certain socio-demographic and COVID-19-related variables played a significant role in shaping these findings. Specifically, correlation analysis indicated that females exhibited a higher level of COVID-19 anxiety, which is consistent with existing literature on the subject. Specifically, a recent meta-analysis [55] has highlighted that females reported a higher level of COVID-19 anxiety and fear.

Moreover, younger individuals exhibited higher levels of COVID-19 anxiety in comparison to older individuals. Although this finding appears counterintuitive considering the higher mortality rate associated with older age groups, it has been consistently reported in various studies (e.g., [56]). One possible speculation regarding this result is that older individuals may show lower anxiety levels in response to the COVID-19 pandemic due to their prior experiences with other significant life events. It is conceivable that having encountered and coped with previous critical situations throughout their lives, older people may become less distressed about the prospect of death and have developed more effective coping mechanisms [57].

The interpretation of the positive relationship between having contracted COVID-19 in the past 30 days and education level is intricate. It is plausible that this relationship is merely coincidental. However, considering this finding from a speculative perspective, it is conceivable to hypothesize that individuals with higher levels of education may allocate more resources toward getting tested for potential infection. Nevertheless, it is essential to acknowledge that this is a conjecture and further research should thoroughly investigate this association, ruling out the possibility of a mere chance correlation. Future studies should aim to explore this link in-depth to gain a better understanding of its underlying mechanisms and significance.

Males and older individuals have been found to exhibit higher levels of resilience. While these findings are not universally consistent across the literature, they have been reported in numerous studies investigating the association between gender (e.g., [58]) and age (e.g., [59]) with resilience.

The association between COVID-19 vaccination and higher-reported COVID-19 anxiety may initially appear surprising. However, this relationship can be elucidated by considering that individuals who choose to receive the COVID-19 vaccine may do so due to their heightened anxiety about contracting the virus. Consequently, the vaccination may not fully alleviate this underlying anxiety. A relevant study [60] supports this notion, indicating that individuals who received only one dose of the COVID-19 vaccine experienced less anxiety compared to those who were still awaiting completion of the vaccination cycle. Nevertheless, individuals expressing the lowest level of anxiety were those who had decided not to get vaccinated and had no inclination to do so. This observation aligns with the low levels of anxiety reported in the results section for participants who opted not to receive the vaccine.

Importantly, the study found that resilience mediated the relationship between the locomotion regulatory mode and COVID-19 anxiety. This means that the positive relationship between the locomotion regulatory mode and resilience partially explains why people with a goal-oriented approach to life are less likely to experience anxiety related to the pandemic. In other words, individuals high on locomotion for their nature display a high level of internal resources, are proactive and maintain a steadfast focus on their goals. Hence, they are more likely to be resilient and this helps them better cope with anxiety caused by the pandemic.

Our findings are consistent with prior research showing an association between locomotion and positive well-being [33,35]. They are also in line with previous works that have demonstrated how personality and motivational dispositions can predict a maladaptive adjustment to stressors in both personal and professional life [61,62]. In this vein, it has been suggested that certain types of individuals are more resilient to chronic stress than others, and locomotion seems to play a fundamental role in this process [47].

Therefore, the present findings provide a unique insight into the basic mechanisms of the locomotion regulatory mode by revealing its direct and indirect effects on COVID-19 anxiety. Specifically, this work focuses on the protective role of locomotion, helping understand how individuals who exhibit locomotion tendencies are equipped to cope with the challenges posed by the COVID-19 pandemic. With this regard, understanding factors that influence people’s ability to cope with the pandemic can help interventions and support strategies to promote resilience and mental health during these challenging times. 

Specifically, the peculiarity of locomotion orientation points to the importance of targeting these aspects to alleviate the harmful psychological consequences of COVID-19 pandemic. Since the locomotion regulatory mode can be situationally induced [63,64], it may be possible to design health campaigns oriented at inducing locomotion motivation. This approach could be helpful in taking advantage of locomotion strengths to sustain people in finding more adaptive ways to manage very difficult circumstances (such as a pandemic). Such campaigns would emphasize pursuing goals, empowerment and proactive behaviors, aligning with the characteristics of the locomotion mode. Strategies may include setting specific health-related goals, taking initiative and highlighting positive outcomes of healthy behaviors. Social support and community engagement would be integrated to foster a sense of belonging, and reinforce healthy habits [65]. By adopting this approach, individuals can take control of their health despite difficult circumstances, leading to improved well-being and resilience across diverse populations.

Several factors limit the generalizability of these findings. Specifically, the first aspect is related to the cross-sectional nature of the research that impedes longitudinal assumption, only allowing for the testing of relationships between predictors and outcomes at a specific point in time. Furthermore, the use of a convenience sample should be reported (i.e., recruited using snowball sampling), along with the imbalance of the sample (i.e., 82% female). Another limitation pertains to the use of self-report measures. In fact, data derived from the same source are potentially subject to common method/source bias. For this reason, it would be beneficial to have, for instance, a physiological measure of anxiety.

Additionally, it should be addressed that such data are limited to responses from individuals living in Italy during a specific phase of the COVID-19 pandemic. With this regard, it is important to note that many aspects of life in Italy have changed during the pandemic. In this vein, we could not be sure that the pattern of our results could have been similar in the first phase of the pandemic. Consequently, the findings of the present study exhibit limited generalizability to other populations and contexts, emphasizing the need for caution when extrapolating the results beyond the specific sample and conditions examined. Several factors contribute to this limited generalizability. First, the study participants may not be representative of the broader population due to potential sampling biases. Second, as mentioned, the context in which the research was conducted may have unique characteristics that could influence the observed outcomes. Third, cultural, socio-economic and environmental differences in various settings may significantly impact the applicability of the results to diverse populations.

To address these limitations, future studies should aim to replicate the findings in different settings and among more diverse groups, encompassing various demographics and cultural backgrounds. A broader range of participants and contexts will enhance the external validity of the research and contribute to a more comprehensive understanding of the phenomenon under investigation.

As previously reported, data were collected using a web-mediated survey. Although such a procedure may lead to self-selected and potentially biased samples [66], in the case of COVID-19 and in the presence of a scenario changing very quickly, it allows for collecting data with an extreme rapidity. With this regard, it has also been showed that psychometric properties of online surveys seem to be equivalent to those of paper-and-pencil formats in terms of both internal validity and reliability [66,67].

Nevertheless, another limitation of the present research pertains to the lack of assessment mode measure, which do not allow for identifying its effects on COVID-19 anxiety and the possible interaction effects with the locomotion mode. With this regard, a recent work [68] has highlighted the detrimental impact of the assessment regulatory mode vulnerabilities in response to the COVID-19 pandemic. Specifically, these vulnerabilities were found to be indirectly linked to heightened psychological distress manifested through fear of missing out, challenges in engaging in activities and involvement in negative behaviors. Hence, future studies should do well to examine the relationship between both regulatory modes, resilience and COVID-19 anxiety. In light of the substantial negative association documented in the literature between assessment and psychological states (e.g., [37]), it is conceivable to posit that a particular combination of traits, specifically, high locomotion coupled with low assessment, could potentially exert a preventive influence on COVID-19 anxiety. The inclination toward proactive behaviors and dynamic responses (high locomotion) may act as a protective factor against anxiety, while concurrently adopting a reduced focus on evaluating potential threats and negative information (low assessment) may further contribute to anxiety reduction [69].

Longitudinal studies are necessary to comprehensively investigate the enduring impact of locomotion on resilience and anxiety. While the present research provides valuable insights into these associations, a longitudinal approach allows for the examination of changes and developments over an extended period. Understanding how locomotion influences resilience and anxiety over time is vital as it offers the opportunity to uncover potential patterns, trends and causal relationships. 

Moreover, such longitudinal investigations can shed light on critical factors that may moderate or mediate the relationship between locomotion and psychological outcomes. While the present research suggests that resilience plays a crucial role in mitigating anxiety during challenging circumstances, exploring the presence of other potential mediators beyond resilience could offer valuable insights. For instance, based on previous research [70], which has demonstrated a positive association between locomotion and grit, one can reasonably speculate that grit, considering its negative association with anxiety [71], could potentially act as a mediator in the relationship between locomotion and COVID-19 anxiety. Another avenue worth exploring is the examination of additional factors that have been recognized as predictors of resilience, such as reduced adversity experienced within the parent/guardian relationship and possessing an internal locus of control [72].

Therefore, future research would benefit from exploring additional factors that could potentially influence the interplay between the locomotion regulatory mode, resilience and COVID-19 anxiety. Conducting such investigations can contribute to a more comprehensive understanding of the underlying psychological mechanisms at play in this context.

## 5. Conclusions

In conclusion, this work provides valuable insights into the relationship between the locomotion regulatory mode, resilience and COVID-19 anxiety. The findings suggest that resilience mediates the relationship between locomotion and anxiety, highlighting the importance of being resilient when coping with worry and concern caused by the pandemic. These findings have important implications for mental health interventions and support strategies during the COVID-19 pandemic and beyond.

## Figures and Tables

**Figure 1 ijerph-20-06533-f001:**
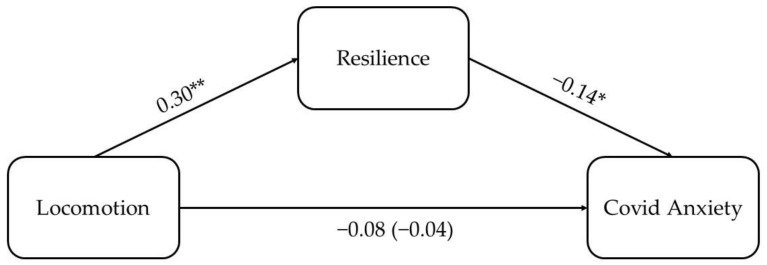
Mediation model. Note: * *p* < 0.05; ** *p* < 0.01. Standardized regression coefficients (B) are reported. The regression coefficient of the predictor when the mediator was included in the model is shown in brackets.

**Table 1 ijerph-20-06533-t001:** Sample characteristics.

	Participants (*n* = 243)
Gender	
Female	199 (81.9%)
Male	44 (19.1%)
Diverse	-
Age distribution	
18–25	102 (42%)
26–35	67 (27.5%)
36–45	41 (16.9%)
>45	33 (13.6%)
Type of school	
Primary school diploma	-
Middle school diploma	1 (0.4%)
High school degree	134 (55.1%)
University degree	92 (37.9%)
Post-graduate specialization	16 (6.6%)
Vaccination status	
Yes	230
No	13

**Table 2 ijerph-20-06533-t002:** Correlations between variables.

	1	2	3	4	5	6	7	8
1. Gender	(-)							
2. Age	0.04							
3. Education	−0.09	0.26 **						
4. COVID-19 Infection	−0.09	−0.08	0.15 *					
5. COVID-19 Vaccination	0.02	0.05	−0.01	0.16 *				
6. Locomotion	−0.14*	0.16 *	0.10	−0.02	0.08			
7. Resilience	0.10	0.23 **	0.04	0.01	0.04	0.31 **		
8. COVID-19 Anxiety	−0.21 **	−0.15 *	−0.03	0.11	0.14 *	−0.06	−0.18 **	(-)

Note: ** *p* ≤ 0.01; * *p* ≤ 0.05.

**Table 3 ijerph-20-06533-t003:** Summary of the regression models.

	Resilience			COVID-19 Anxiety		
	*B*	*SE*	*p*	*B*	*SE*	*p*
Gender	0.27	0.13	0.035	−0.33	0.11	0.003
Age	0.01	0.00	0.002	−0.01	0.01	0.285
Education	−0.04	0.08	0.586	−0.04	0.07	0.585
COVID-19 Infection	0.09	0.14	0.522	0.21	0.12	0.081
COVID-19 Vaccination	0.12	0.22	0.573	0.48	0.19	0.011
Locomotion	0.38	0.08	0.000	−0.04	0.07	0.558
Resilience				−0.12	0.06	0.038
R^2^		0.15			0.11	

## Data Availability

The data presented in this study are available on request from the corresponding author.

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
