# Peer review of "Effects of Locomotion Regulatory Mode on COVID-19 Anxiety: The Mediating Role of Resilience"

_ijerph, 2023, doi:10.3390/ijerph20156533_

Round 1
Reviewer 1 Report
This manuscript provides a comprehensive overview of the relationship between locomotion regulatory mode, resilience, and COVID-19 anxiety. However, there are a few areas where the manuscript could be further improved. Here are some suggestions:
Introduction:
1.Streamline the subsections: Consider streamlining the subsections within the introduction. For example, merge subsections 1.1 (COVID-19 anxiety predictors) and 1.2 (Resilience and anxiety) to provide a more coherent and focused discussion on the factors influencing anxiety during the pandemic.
2.Clarify the gap in the literature: Clearly articulate the gap in the literature that the study aims to address. Highlight the limited research on the role of regulatory modes, specifically locomotion, in the context of COVID-19 anxiety and resilience. Explain why this gap is important to fill and how the current study contributes to filling this gap.
3.Provide a more focused rationale for studying locomotion regulatory mode: Expand on the rationale for focusing on locomotion regulatory mode. Explain why understanding locomotion is relevant in the context of the COVID-19 pandemic. Discuss how locomotion, with its emphasis on goal-directed action and progress, may influence individuals' ability to cope with challenges and reduce anxiety related to the pandemic.
4. Expand on the relationship between resilience and anxiety: Provide a more detailed overview of the relationship between resilience and anxiety. Discuss the theoretical underpinnings of this relationship, such as how resilience contributes to adaptive coping and buffers against the negative impact of stressors. Include additional references to support this discussion and highlight the significance of studying resilience in the context of COVID-19 anxiety.
Methods:
1.The specific content of data analysis is missing in the method section: for example, what kind of regression model is used? and which model is used in mediation analysis?
2.Clarify the recruitment process: Provide more details about how participants were recruited using snowball sampling. Explain the initial participants' selection process and how they were encouraged to refer others to participate. Additionally, mention any inclusion or exclusion criteria used to select participants.
3. Expand on the socio-demographic data: Provide a more comprehensive description of the socio-demographic variables collected. Include information such as participants' distribution across age ranges, and any other relevant demographic characteristics. This will help readers understand the sample composition and potential implications for generalizability.
4. Discuss the translation and adaptation of measures: If the scales were translated or adapted for use in the current study, provide information about the translation process, any validation procedures conducted, and the reliability of the translated scales. This information is particularly relevant for the Italian version of the Locomotion Scale.
5. Provide information on data collection: Describe how data were collected, including the platform or software used for the web-mediated survey. Provide details on the duration of the survey, any instructions or guidance given to participants, and the anonymity and confidentiality measures in place to protect participants' data.
Results:
1.Clarify the sample characteristics: provide a table with more descriptive statistics details about the participants’ characteristics, such as age range, gender distribution, educational background, vaccination status, anxiety level, and any other relevant demographic information.
2. Discuss the control variables: In the regression analysis, discuss the control variables (gender, age, education, COVID-19 vaccination, and COVID-19 infection) in more detail. Explain why these variables were included and how they may have influenced the relationship between locomotion, resilience, and COVID-19 anxiety. Discuss the significance or non-significance of these control variables in relation to the research hypotheses.
Discussion:
1. Clarify the unique contribution of the study: Clearly state how the present findings contribute to the existing literature. Highlight the novelty and unique insights provided by examining the role of locomotion regulatory mode in the context of COVID-19 anxiety. Emphasize why this understanding is important and how it extends our knowledge of coping mechanisms during challenging times.
2. Expand on the practical implications: Discuss the practical implications of the study's findings in more detail. Elaborate on how interventions and support strategies could be designed to promote resilience and mental health based on the understanding of locomotion regulatory mode. Provide specific recommendations or ideas for potential interventions that could target locomotion tendencies.
3.Discuss the implications for public health campaigns: Further explore the suggestion of designing health campaigns to induce locomotion motivation. Discuss how such campaigns could be structured, what strategies could be employed, and how they could be effective in helping individuals cope with challenging circumstances, including pandemics.
4. Address the limitations more thoroughly: Provide a more detailed discussion of the limitations mentioned, such as the cross-sectional nature of the research, convenience sampling, and self-report measures. Explain how these limitations may have affected the study's findings and discuss their implications for the generalizability of the results. Additionally, suggest potential solutions or alternative approaches that could be employed in future research to overcome these limitations. For example, discuss the need for longitudinal studies to explore the long-term effects of locomotion on resilience and anxiety over time.
5. Consider alternative explanations and future research directions: Discuss alternative explanations for the findings and acknowledge potential confounding variables or factors that were not accounted for in the current study. This will demonstrate a critical evaluation of the research and open up possibilities for future investigations. Propose specific research directions that could build upon the current study, such as examining the interaction effects between locomotion and assessment regulatory modes in relation to COVID-19 anxiety. Additionally, consider suggesting investigations into other potential factors that may moderate or mediate the relationship between locomotion, resilience, and COVID-19 anxiety.
6. Address the external validity: Given that the data were collected from individuals living in Italy during a specific phase of the COVID-19 pandemic, discuss the potential impact of cultural, social, and contextual factors on the findings. Consider the generalizability of the results to other populations and contexts, and highlight the need for future studies to replicate the findings in diverse settings.
Overall, the English quality of this article is good.
Author Response
Dear Reviewer,
Thank you very much for your kind remarks about our paper, entitled “Effects of Locomotion Regulatory Mode on COVID-19 Anxiety: The Mediating Role of Resilience”, and for your helpful comments. We have now carried out a revision of the manuscript in response to those comments and are re-submitting it for your consideration. We found the commentary on the previous version of the paper very useful and thought-provoking, and the present version of the paper constitutes a careful reworking of the prior submission. Let me now indicate how we responded to each of your comments on the prior version of the paper.
Introduction:
1.Streamline the subsections: Consider streamlining the subsections within the introduction. For example, merge subsections 1.1 (COVID-19 anxiety predictors) and 1.2 (Resilience and anxiety) to provide a more coherent and focused discussion on the factors influencing anxiety during the pandemic.
We addressed this issue by combining the two sections to create a single subsection in which we described the main predictors of Covid-19 anxiety (p. 2).
2.Clarify the gap in the literature: Clearly articulate the gap in the literature that the study aims to address. Highlight the limited research on the role of regulatory modes, specifically locomotion, in the context of COVID-19 anxiety and resilience. Explain why this gap is important to fill and how the current study contributes to filling this gap.
We created a new subsection, “1.5 Research Hypotheses”, in which we have addressed the issue by explicitly clarifying the gap in the literature pertaining to the role of regulatory modes, particularly locomotion, in relation to COVID-19 anxiety and resilience. By emphasizing the scarcity of research in this specific area, we underscored the significance of filling this gap and highlighted how our study contributes to advancing the understanding of COVID-19's psychological impact and coping mechanisms.
- Provide a more focused rationale for studying locomotion regulatory mode: Expand on the rationale for focusing on locomotion regulatory mode. Explain why understanding locomotion is relevant in the context of the COVID-19 pandemic. Discuss how locomotion, with its emphasis on goal-directed action and progress, may influence individuals' ability to cope with challenges and reduce anxiety related to the pandemic.
We have addressed this issue, in the research hypotheses section, by highlighting locomotion's emphasis on goal-directed action and progress. We have underlined how it can significantly influence individuals' coping abilities and potentially mitigate anxiety associated with the challenges posed by the pandemic.
- Expand on the relationship between resilience and anxiety: Provide a more detailed overview of the relationship between resilience and anxiety. Discuss the theoretical underpinnings of this relationship, such as how resilience contributes to adaptive coping and buffers against the negative impact of stressors. Include additional references to support this discussion and highlight the significance of studying resilience in the context of COVID-19 anxiety.
We have addressed this issue by delving into a comprehensive overview of the relationship between resilience and anxiety. Additionally, we bolstered this discussion by incorporating additional references to support the significance of studying resilience, particularly in the context of COVID-19 anxiety, thus providing a robust understanding of the interplay between these psychological constructs (p. 2-3).
Methods:
1.The specific content of data analysis is missing in the method section: for example, what kind of regression model is used? and which model is used in mediation analysis?
In section 3.2 (Regression model), we have specified the regression model used for the mediation analysis, as well as the procedure for the bootstrap samples.
2. Clarify the recruitment process: Provide more details about how participants were recruited using snowball sampling. Explain the initial participants' selection process and how they were encouraged to refer others to participate. Additionally, mention any inclusion or exclusion criteria used to select participants.
We have addressed this issue, in the method section, by providing a clarification of the recruitment process. Furthermore, we included information on any specific inclusion or exclusion criteria utilized in the participant selection process.
- Expand on the socio-demographic data: Provide a more comprehensive description of the socio-demographic variables collected. Include information such as participants' distribution across age ranges, and any other relevant demographic characteristics. This will help readers understand the sample composition and potential implications for generalizability.
We have addressed this issue by presenting a more comprehensive description of the socio-demographic variables collected, including a detailed distribution of participants across various age ranges and other relevant demographic characteristics (Table 1). This enriched presentation helps readers gain a better understanding of the sample composition.
- Discuss the translation and adaptation of measures: If the scales were translated or adapted for use in the current study, provide information about the translation process, any validation procedures conducted, and the reliability of the translated scales. This information is particularly relevant for the Italian version of the Locomotion Scale.
We have addressed this issue by discussing the translation and adaptation of measures, specifically focusing on the Italian version of the Locomotion Scale (p. 5).
- Provide information on data collection: Describe how data were collected, including the platform or software used for the web-mediated survey. Provide details on the duration of the survey, any instructions or guidance given to participants, and the anonymity and confidentiality measures in place to protect participants' data.
We have addressed this issue by thoroughly providing information on data collection, elucidating the platform used for the web-mediated survey. We described the survey's duration, provided instructions and guidance given to participants, and outlined the anonymity and confidentiality measures implemented to safeguard participants' data (p. 4).
Results:
- Clarify the sample characteristics: provide a table with more descriptive statistics details about the participants’ characteristics, such as age range, gender distribution, educational background, vaccination status, anxiety level, and any other relevant demographic information.
By offering a more comprehensive description of the collected socio-demographic variables, which includes a detailed distribution of participants across different age ranges and relevant demographic characteristics (Table 1, p. 4-5), we have enhanced the presentation to aid readers in gaining a deeper understanding of the sample composition.
- Discuss the control variables: In the regression analysis, discuss the control variables (gender, age, education, COVID-19 vaccination, and COVID-19 infection) in more detail. Explain why these variables were included and how they may have influenced the relationship between locomotion, resilience, and COVID-19 anxiety. Discuss the significance or non-significance of these control variables in relation to the research hypotheses.
We have addressed this issue by providing a detailed discussion of the control variables (gender, age, education, COVID-19 vaccination, and COVID-19 infection) both in the correlation (p. 6) and in the regression analysis (p.7). We explained the rationale behind including these variables and how they may have influenced the relationship between locomotion, resilience, and COVID-19 anxiety. Additionally, we thoroughly discussed the significance or non-significance of these control variables in relation to the research hypotheses, offering valuable insights into their potential impact on the study's outcomes (p.8-9).
Discussion:
- Clarify the unique contribution of the study: Clearly state how the present findings contribute to the existing literature. Highlight the novelty and unique insights provided by examining the role of locomotion regulatory mode in the context of COVID-19 anxiety. Emphasize why this understanding is important and how it extends our knowledge of coping mechanisms during challenging times.
We have addressed this issue by explicitly stating the study's unique contribution to the existing literature, emphasizing the novel insights gained from investigating the role of locomotion regulatory mode in the context of COVID-19 anxiety. We highlighted the importance of this understanding and how it extends our knowledge of coping mechanisms during challenging times, ultimately adding valuable information to the field of psychological research in response to the pandemic (p. 8-9).
- Expand on the practical implications: Discuss the practical implications of the study's findings in more detail. Elaborate on how interventions and support strategies could be designed to promote resilience and mental health based on the understanding of locomotion regulatory mode. Provide specific recommendations or ideas for potential interventions that could target locomotion tendencies.
We have addressed this issue by providing a more detailed discussion of the practical implications arising from the study's findings. We elaborated on how the understanding of locomotion regulatory mode can inform the design of interventions and support strategies to promote resilience and mental health (p. 9).
3. Discuss the implications for public health campaigns: Further explore the suggestion of designing health campaigns to induce locomotion motivation. Discuss how such campaigns could be structured, what strategies could be employed, and how they could be effective in helping individuals cope with challenging circumstances, including pandemics.
We have addressed this issue by thoroughly discussing the implications for public health campaigns, particularly exploring the idea of designing campaigns to induce locomotion orientation. We delved into how such campaigns could be structured, the strategies that could be employed, and how they could effectively assist individuals in coping with challenging circumstances, including pandemics (p. 9)
- Address the limitations more thoroughly: Provide a more detailed discussion of the limitations mentioned, such as the cross-sectional nature of the research, convenience sampling, and self-report measures. Explain how these limitations may have affected the study's findings and discuss their implications for the generalizability of the results. Additionally, suggest potential solutions or alternative approaches that could be employed in future research to overcome these limitations. For example, discuss the need for longitudinal studies to explore the long-term effects of locomotion on resilience and anxiety over time.
We have addressed this issue by offering a comprehensive and in-depth discussion of the limitations mentioned in the study, including the cross-sectional design, convenience sampling, and reliance on self-report measures. We explained how these limitations might have influenced the study's findings and discussed their implications for the generalizability of the results (p. 9-10). Moreover, we provided potential solutions and alternative approaches for future research, highlighting the importance of conducting longitudinal studies to investigate the long-term effects of locomotion on resilience and anxiety over time (p. 10).
- Consider alternative explanations and future research directions: Discuss alternative explanations for the findings and acknowledge potential confounding variables or factors that were not accounted for in the current study. This will demonstrate a critical evaluation of the research and open up possibilities for future investigations. Propose specific research directions that could build upon the current study, such as examining the interaction effects between locomotion and assessment regulatory modes in relation to COVID-19 anxiety. Additionally, consider suggesting investigations into other potential factors that may moderate or mediate the relationship between locomotion, resilience, and COVID-19 anxiety.
We have addressed this issue by engaging in a thorough discussion of alternative explanations for the findings and acknowledging potential confounding variables or factors that were not addressed in the current study, showcasing a critical evaluation of the research. Furthermore, we proposed specific research directions that could expand upon the current study, such as exploring the interaction effects between locomotion and assessment regulatory modes in relation to COVID-19 anxiety, and suggesting investigations into other potential factors that may moderate or mediate (e.g., grit) the relationship between locomotion, resilience, and COVID-19 anxiety (p. 10).
- Address the external validity: Given that the data were collected from individuals living in Italy during a specific phase of the COVID-19 pandemic, discuss the potential impact of cultural, social, and contextual factors on the findings. Consider the generalizability of the results to other populations and contexts, and highlight the need for future studies to replicate the findings in diverse settings.
We have addressed this issue by thoroughly discussing the potential impact of cultural, social, and contextual factors on the findings, considering that the data were collected from individuals in Italy during a specific phase of the COVID-19 pandemic. We also emphasized the importance of considering the generalizability of the results to other populations and contexts, and highlighted the necessity for future studies to replicate the findings in diverse settings, thus addressing the issue of external validity and promoting a more comprehensive understanding of the broader implications of the research (p. 9-10).
Reviewer 2 Report
The study highlights the importance of considering both locomotion and resilience when managing anxiety related to COVID-19. It suggests that interventions aimed at enhancing resilience could be particularly beneficial for individuals with low locomotion regulatory mode. Overall, the research sheds light on potential factors that can influence individuals' responses to the COVID-19 pandemic and its impact on mental health.
1. What are the main study variables examined in Table 1?
2. Could you explain the correlation between locomotion and resilience mentioned in Table 1?
3. How is COVID-19 anxiety related to resilience and gender according to Table 1?
4. What regression model was used to test the relationship between locomotion, resilience, and COVID-19 anxiety, and how was it analyzed?
5. Could you elaborate on the control variables that were entered into the regression model?
6. What were the main findings of the regression analysis as reported in Table 2?
7. In Figure 1, what does the mediation model depict, and what does it tell us about the relationship between locomotion, resilience, and COVID-19 anxiety?
8. What implications do the study's findings have for mental health interventions during the COVID-19 pandemic?
9. What are the limitations of the study that might affect the generalizability of the findings?
10. Are there any recommendations for future research based on the results of this study?
The content uses mostly clear and concise sentence structures, which is good for readability. However, in a few instances, there are longer sentences that could be broken down for better clarity.
Author Response
Dear Reviewer,
Thank you very much for your kind remarks about our paper, entitled “Effects of Locomotion Regulatory Mode on COVID-19 Anxiety: The Mediating Role of Resilience”, and for your helpful comments. We have now carried out a revision of the manuscript in response to those comments and are re-submitting it for your consideration. We found the commentary on the previous version of the paper very useful and thought-provoking, and the present version of the paper constitutes a careful reworking of the prior submission. Let me now indicate how we responded to each of your comments on the prior version of the paper.
- What are the main study variables examined in Table 1?
We have incorporated a new table to provide a more detailed description of the socio-demographic variables characterizing the sample. In addition, commenting the results of the correlation table (now Table 2), we have clarified the role of control variables and main variables included in the study (p. 6).
- Could you explain the correlation between locomotion and resilience mentioned in Table 1?
We have addressed this issue by providing a clear explanation of the correlation between locomotion and resilience, as mentioned in Table 2. We elaborated on the statistical relationship between these variables, discussing the strength and direction of the correlation, which allows readers to better understand the association between locomotion and resilience in the context of the study.
- How is COVID-19 anxiety related to resilience and gender according to Table 1?
We have addressed this issue by providing a comprehensive explanation of the relationship between COVID-19 anxiety, resilience, and gender as depicted in Table 2. We discussed the specific correlations between these variables, elucidating how they are interrelated and providing insights into the associations between COVID-19 anxiety, resilience, and gender within the study's context.
- What regression model was used to test the relationship between locomotion, resilience, and COVID-19 anxiety, and how was it analyzed?
In section 3.2 (Regression model), we have specified the regression model used for the mediation analysis, as well as the procedure for the bootstrap samples.
- Could you elaborate on the control variables that were entered into the regression model?
We have addressed this issue by providing a detailed discussion of the control variables (gender, age, education, COVID-19 vaccination, and COVID-19 infection) both in the correlation (p. 6) and in the regression analysis (p.7). We explained the rationale behind including these variables and how they may have influenced the relationship between locomotion, resilience, and COVID-19 anxiety. Additionally, we thoroughly discussed the significance or non-significance of these control variables in relation to the research hypotheses, offering valuable insights into their potential impact on the study's outcomes (p.8-9).
- What were the main findings of the regression analysis as reported in Table 2?
We have addressed this issue by providing a detailed discussion of the results presented in Table 3, which encompasses an in-depth analysis of the effects and significance of both control and main variables incorporated in the study (p. 7). Through this comprehensive examination, we shed light on how each variable contributes to the overall understanding of the research outcomes, thereby enriching the interpretation and implications drawn from the study (p. 8-9).
- In Figure 1, what does the mediation model depict, and what does it tell us about the relationship between locomotion, resilience, and COVID-19 anxiety?
We have addressed this issue by providing a clear explanation of the mediation model depicted in Figure 1. This model illustrates how locomotion influences COVID-19 anxiety through its impact on resilience, offering valuable insights into the indirect relationship between locomotion, resilience, and COVID-19 anxiety (p. 7). Through this mediation analysis, we gain a deeper understanding of the underlying mechanisms that link these psychological constructs and their implications for coping with pandemic-related anxiety.
- What implications do the study's findings have for mental health interventions during the COVID-19 pandemic?
We have addressed this issue by thoroughly discussing the implications for public health campaigns, particularly exploring the idea of designing campaigns to induce locomotion orientation. We delved into how such campaigns could be structured, the strategies that could be employed, and how they could effectively assist individuals in coping with challenging circumstances, including pandemics (p. 9)
- What are the limitations of the study that might affect the generalizability of the findings?
We have addressed this issue by offering a comprehensive and in-depth discussion of the limitations mentioned in the study, including the cross-sectional design, convenience sampling, and reliance on self-report measures. We explained how these limitations might have influenced the study's findings and discussed their implications for the generalizability of the results (p. 9-10).
- Are there any recommendations for future research based on the results of this study?
We have addressed this issue by proposing specific research directions that could expand upon the current study, such as exploring the interaction effects between locomotion and assessment regulatory modes in relation to COVID-19 anxiety, and suggesting investigations into other potential factors that may moderate or mediate (e.g., grit) the relationship between locomotion, resilience, and COVID-19 anxiety (p. 10). Moreover, we provided potential solutions and alternative approaches for future research, highlighting the importance of conducting longitudinal studies to investigate the long-term effects of locomotion on resilience and anxiety over time (p. 10).